# Reconstruction of the metabolic network of *Pseudomonas aeruginosa* to interrogate virulence factor synthesis

Jennifer A. Bartell[1,2,*], Anna S. Blazier[2,*], Phillip Yen[2], Juliane C. Thøgersen[3], Lars Jelsbak[3], Joanna B. Goldberg[4,5] & Jason A. Papin[2]

Virulence-linked pathways in opportunistic pathogens are putative therapeutic targets that may be associated with less potential for resistance than targets in growth-essential pathways. However, efficacy of virulence-linked targets may be affected by the contribution of virulence-related genes to metabolism. We evaluate the complex interrelationships between growth and virulence-linked pathways using a genome-scale metabolic network reconstruction of *Pseudomonas aeruginosa* strain PA14 and an updated, expanded reconstruction of *P. aeruginosa* strain PAO1. The PA14 reconstruction accounts for the activity of 112 virulence-linked genes and virulence factor synthesis pathways that produce 17 unique compounds. We integrate eight published genome-scale mutant screens to validate gene essentiality predictions in rich media, contextualize intra-screen discrepancies and evaluate virulence-linked gene distribution across essentiality datasets. Computational screening further elucidates interconnectivity between inhibition of virulence factor synthesis and growth. Successful validation of selected gene perturbations using PA14 transposon mutants demonstrates the utility of model-driven screening of therapeutic targets.

[1] The Novo Nordisk Foundation Center for Biosustainability, Technical University of Denmark, 2970 Hørsholm, Denmark. [2] Biomedical Engineering, University of Virginia, Charlottesville, Virginia 22908, USA. [3] Department of Biotechnology and Biomedicine, Technical University of Denmark, 2800 Kgs. Lyngby, Denmark. [4] Department of Pediatrics, Division of Pulmonology, Allergy/Immunology, Cystic Fibrosis and Sleep, Children's Healthcare of Atlanta, Atlanta, Georgia 30322, USA. [5] Emory + Children's Center for Cystic Fibrosis Research, Emory University and Children's Healthcare of Atlanta, Atlanta, Georgia 30322, USA. * These authors contributed equally to this work. Correspondence and requests for materials should be addressed to J.A.P. (email: papin@virginia.edu).

There is a need for new drugs that effectively inhibit microbial infection while avoiding the development of resistance. Traditional antibiotics that inhibit growth of bacteria by targeting growth-essential functions actively select for antibiotic-resistant mutants that overtake the infection. This growth-based selection promotes the rapid development of resistance and consequently exacerbates infections[1], resulting in substantial patient morbidity, mortality and health-care costs[2]. Inhibiting mechanisms of infection by targeting the synthesis of virulence factors (VFs) and virulence-linked genes may be a promising new therapeutic strategy that avoids growth-based target selection, improves patient outcomes and mitigates the spread of resistance[1,3,4]. However, genes that contribute to growth and genes that contribute to virulence are not necessarily distinct actors in an organism's genetic network; understanding the impact of genes on each pathogen directive (growth versus virulence) is critical to therapy design and prediction of resistance development.

Virulence-linked genes contribute to survival and fitness within a host. Many of these genes encode the synthesis pathways of VFs, pathogen-produced small molecules that are involved in activities such as iron sequestration and bacterial communication that enable adaptation to the host environment and enhance infection potential[5,6]. In targeting the synthesis of these metabolites, resistance may develop more slowly because of weakened selection pressure versus traditional targets that directly impact growth-essential catabolism of substrates or cell wall construction and repair[3]. However, our understanding of the role of virulence-linked genes is evolving[7]—significant links between virulence and pathogen metabolism are now emerging. For example, antibiotic pigments called phenazines enable opportunistic bacteria to combat the effects of immune cell oxidative bursts, but these pigments may also induce rewiring of redox-linked pathways within the pathogen[8]. Furthermore, the production of virulence-linked compounds relies on essential components of central metabolism that connect substrate catabolism to VF synthesis pathways. A clear division between therapeutic targets impacting growth and virulence is therefore not straightforward[7]. We need to map the interconnectivity of these systems to identify genes that contribute to either or both systems, determine their function and essentiality in a clinically relevant environment, and estimate the impact of their inhibition on virulence versus growth.

To study the relationship between VF synthesis and growth from a systems level perspective, we used genome-scale metabolic network models (GEMs). Assembled from annotated genomic data, GEMs are mathematical frameworks that incorporate biochemical, genetic and cell phenotypic data and account for hundreds to thousands of gene-protein-reaction (GPR) relationships and reaction stoichiometry and directionality[9]; they have been used to predict novel drug targets that inhibit growth[10] as well as probe the capability of an organism to synthesize various metabolites[11], including VFs (ref. 12).

Here, we present a new GEM of *Pseudomonas aeruginosa* strain PA14 (iPau1129) as well as an updated GEM of reference strain *P. aeruginosa* PAO1 (iPae1146). *P. aeruginosa* is a Gram-negative opportunistic pathogen capable of developing multi-drug antibiotic resistance, hospital-acquired infections[13–15], and infections in cystic fibrosis patient lungs, burn wounds and immunocompromised individuals. We validate our GEMs using substrate utilization data and gene essentiality screens from transposon mutant libraries and use six previously published transposon sequencing (Tn-seq) screens to evaluate essential virulence-linked genes[16–18]. To study the relationship between VF production and growth, we compare the effect of *in silico* gene knockouts on synthesis of biomass versus 17 VFs and

identified genes uniquely critical for VF production, genes solely important for the synthesis of biomass, as well as genes involved in both VF production and biomass production. A case study of the VF pyoverdine shows the utility of GEMs in probing network dependencies that offer novel insights into links between virulence and metabolism that may enhance design of cycled or combination drug therapies as well as reduce the development of resistance.

## Results

**Metabolic network reconstruction of *P. aeruginosa*.** Here, we present an updated GEM of *P. aeruginosa* strain PAO1 (iPae1146) as well as a new GEM of *P. aeruginosa* strain PA14 (iPau1129) (for ease of reference in this study, we refer to these reconstructions as mPAO1 and mPA14, respectively). The network reconstruction process began with previous *P. aeruginosa* PAO1 GEMs (refs 19,20). We implemented a more detailed biomass equation, incorporated new biological information, and curated the model against carbon source utilization and gene essentiality data (see below). We also assigned potential roles to 59 and 44 genes annotated as hypothetical proteins in PAO1 and PA14 genome annotations from the Pseudomonas Genome Database (PGD), respectively. In conclusion, the new GEM mPA14 accounts for the function of 1,129 genes, 1,495 reactions and 1,286 metabolites while the updated GEM mPAO1 accounts for the function of 1,146 genes, 1,493 reactions and 1,284 metabolites (Fig. 1a). The distribution of genes, metabolites and reactions in mPA14 across a variety of KEGG functional categories is shown in Fig. 1b (for the distribution of mPAO1, see Supplementary Fig. 1).

During curation, we specifically accounted for the synthesis pathways of several small molecule VFs. *P. aeruginosa* produces an array of VFs which can be grouped into several categories including exopolysaccharides, lipopolysaccharides, phenazines, quorum sensing signal molecules, siderophores and surfactants[21,22]. Table 1 lists the compounds that can be synthesized by mPA14—the six italicized factors are new to mPAO1 and mPA14 compared to previous GEMs. Bolded dihydroaeruginoic acid is a recently identified PA14-specific VF included only in mPA14 (ref. 23). Additionally, we evaluated a list of 454 genes linked to virulence of PAO1, PA14 or both in the Virulence Factor Annotations tool from the recently updated PGD to identify model genes that are associated with virulence. Only 123 of these virulence-linked genes were annotated as part of a BRITE metabolic pathway by KEGG, and 49 of the 454 genes were annotated as hypothetical proteins. Using KEGG and PseudoCAP annotations (functional system annotations developed by the PGD) as well as literature on VF synthesis, we focused on accounting for genes relevant to metabolism and virulence-linked synthesis pathways. Ultimately, there are 112 and 108 virulence-linked genes incorporated into mPAO1 and mPA14, respectively[24].

**Model validation.** We used two data types to curate and validate the models: a carbon source utilization data set and a published gene essentiality data set. We generated the substrate utilization data set using BIOLOG phenotype microarrays, which indicated whether PAO1 and PA14 were able to grow on particular carbon sources. We then compared these results to model predictions of biomass production (an approximation of growth) on different minimal media. After extensive transport reaction curation and refinement of metabolic pathways, mPA14 and mPAO1 account for 91 and 93 carbon sources and predict utilization with accuracies of 81% and 80%, respectively (Supplementary Fig. 2).

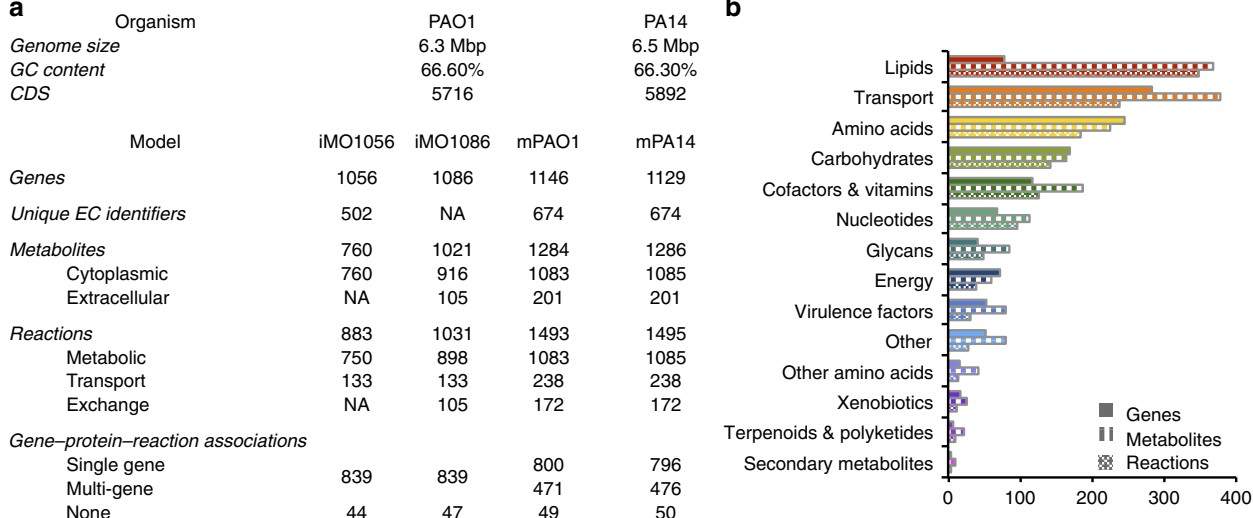

**Figure 1 | Network model characteristics.** (**a**) Properties of the updated PAO1 model as compared to previously published GEMs for *P. aeruginosa*, iMO1056 and iMO1096, as well as properties of the new PA14 model. (**b**) The number of genes, metabolites and reactions in mPA14 grouped into functional categories as defined by KEGG (ref. 74). For the distribution of genes, metabolites and reactions in mPAO1, see Supplementary Fig. 1.

**Table 1 | Small molecules associated with virulence accounted for in mPA14 and mPAO1.**

| Category | Virulence factor | Metabolite ID |
|---|---|---|
| Exopolysaccharide | Alginate | cpd17074[c] |
| Lipopolysaccharide | A-band-O-antigen | cpd17056[c] |
| | B-band-O-antigen | cpd17057[c] |
| | Lipid A | cpd17066[c] |
| Phenazines | Chorismate | cpd00216[c] |
| | *1-Carboxyphenazine* | *cpd17083[c]* |
| | Pyocyanin | cpd01206[c] |
| Quorum sensing signal molecules | Acyl-homoserine-lactone (AHL) | cpd17082[c] and cpd08635[c] |
| | *cis-2-Decenoic acid (DSF)* | *cJB00127[c]* |
| | *2-heptyl-4-quinolone (HHQ)* | *cpd17078[c]* |
| | Pseudomonas quinolone signal (PQS) | cpd17085[c] |
| Siderophores | **Dihydroaeruginoic acid (Dha)** | **cJB00126[c]** |
| | *Pyochelin* | *cpd08828[c]* |
| | *Pyoverdine* | *cPY00164[c]* |
| | *Salicylate* | *cpd00599[c]* |
| Surfactant | Rhamnolipid | cpd17081[c] and cpd17080[c] |

The six italicized factors are new additions to mPA14 and mPAO1 as compared to previous GEMs, and bolded dihydroaeruginoic acid is a recently identified PA14-specific VF included only in mPA14 (ref. 23). The metabolite ID is the compound reference ID used in our models.

For the gene essentiality validation, we used a published data set comprised of the overlap between the essential PAO1 genes identified in Jacobs *et al.*[25] and the essential PA14 genes identified in Liberati *et al.*[26] by creation of transposon insertion mutant libraries in a Luria-Bertani (LB) media background. We used this overlap data set for our curation efforts instead of the individual libraries because there is a higher confidence in which genes are essential since they were not disrupted in either of the PAO1 and PA14 screens and the libraries are validated, publically available and created with established approaches in rich media. To compare this overlap data set with our model predictions, we performed *in silico* single gene knockouts in our models and measured the subsequent effects on biomass production.

Ultimately, both mPA14 and mPAO1 can be used to predict gene essentiality with an accuracy of 91% (Supplementary Fig. 3).

**Virulence associations of Tn-seq-based essential genes.** The recent advent of Tn-seq high-throughput screening has enabled the rapid evaluation of the fitness of a transposon insertion mutant in a given condition. These screens contribute important information regarding bacterial survival in specific contexts. Given our interest in studying the relationship between growth and virulence, we sought to determine how many essential genes were also virulence-linked in recent *P. aeruginosa* Tn-seq screens.

**Table 2 | Comparison of experimental essentiality screens.**

| Strain | PA14 | | PAO1 | | | | | |
|---|---|---|---|---|---|---|---|---|
| Reference<br>Media | Pier<br>LB | Whiteley<br>Sputum | Manoil<br>LB | Whiteley<br>LB | Manoil<br>Sputum | Whiteley<br>Sputum | Manoil<br>Pyruvate | Whiteley<br>Succinate |
| No. of essential genes | 634 | 510 | 201 | 336 | 224 | 445 | 182 | 641 |
| No. of VF-linked essential genes | 49 | 25 | 21 | 20 | 30 | 41 | 27 | 54 |
| %VF-linked essential genes | 7.73 | 4.90 | 10.45 | 5.95 | 13.39 | 9.21 | 14.84 | 8.42 |

The number of essential genes, virulence-linked essential genes (VF-linked essential genes), and the proportion of essential genes that are virulence-linked for all eight of the transposon mutant screens analysed. The data are assembled from studies by the Pier Lab in 2013 (ref. 16) and the Whiteley Lab[17] and Manoil Lab[18] in 2015.

We obtained data from published Tn-seq screens for PAO1 and PA14 in several culturing conditions and identified essential genes for each individual screen. We then compared these individual essential gene lists to a list of either PAO1 or PA14 virulence-linked genes from the PGD to identify virulence-linked essential genes for each screen (Table 2). The number of virulence-linked essential genes identified across the screens ranged from 20 in the Whiteley PAO1 LB data set[17] to 54 in the Whiteley PAO1 succinate data set[17]. Furthermore, the proportion of essential genes that were also virulence-linked varied across the screens, ranging from 4.9% in the Whiteley PA14 sputum data set[17] to 14.8% in the Manoil PAO1 pyruvate data set[18]. This variability in the percentage of virulence-linked essential genes may stem from the variability in transposon insertion coverage of the individual screens. While some screens identified over 600 essential genes[16,17], other screens identified less than 200 essential genes[18] in the same media.

The moderate number of virulence-linked genes present in the Tn-seq screens can partially be explained by the lack of host selection pressure in the generation of the mutant libraries and the imperfect replication of *in vivo* growth conditions in *in vitro* studies. Tn-seq screens performed in infection models have demonstrated that mutants unable to synthesize certain VFs are unable to colonize the infection site[27], suggesting that virulence-linked genes may be essential in some contexts, while elsewhere (such as in liquid culture) they are unnecessary for bacterial fitness. However, using only Tn-seq screens that differ by growth media rather than host selection pressure to contextualize virulence-linked gene essentiality still shows that some virulence-linked genes have important, potentially non-virulence related, functions. This analysis indicates that these genes may play a more focused metabolic role in the development of infection or are capable of dual functions linked to both virulence and growth.

To evaluate the potential overlap of virulence-linked genes with growth activity, we first used mPA14 as a framework to compare sets of growth essential genes and virulence-linked genes that have been curated as functionally relevant to metabolic activity using the Whiteley PA14 sputum screen[17]. Figure 2 shows the model reactions linked to 205 genes required for growth of PA14 in sputum (blue), and the 108 PA14 virulence-linked genes from the PGD (red). The overlap between reactions associated with required genes and virulence-linked genes, totalling 21 reactions (11 genes) are linked to a broad array of systems and present at high density in central metabolic pathways, amino acids, lipids and nucleotide metabolism (overlap reactions in purple). Intriguingly, many reactions associated only with growth or virulence group together in the same pathways, which may indicate functional connections even if specific genes are not shared between the distinct gene sets. This analysis supports the need for a mechanistic evaluation of virulence-linked genes in the context of growth.

**Modelling VF production capabilities.** While infection-based Tn-seq screens have demonstrated that mutants incapable of VF synthesis lack the ability to infect, it is challenging to discern whether this occurs due to the inhibited gene's essentiality for the expression of virulence-linked compound(s), essentiality for growth, or essentiality for both[7,28]. To address this gap in knowledge, we employed genome-scale metabolic network modelling. We implemented a medium that mimics the lung of cystic fibrosis patients (synthetic cystic fibrosis medium, SCFM) in order to more closely model *in vivo* conditions[29]. The ability of *P. aeruginosa* to maintain decades-long infections in the lungs of cystic fibrosis patients may be due to both its metabolic adaptability and deployment of an array of VFs, such that pathway interconnectivity may proffer unique metabolic benefits as well as enable resistance to treatment[30]. Using an *in silico* SCFM medium, we performed *in silico* single-gene knockouts and assessed the levels of growth inhibition and VF synthesis inhibition by normalizing the resulting biomass flux and VF flux to wildtype production levels. By repeating this analysis for all 17 VFs in our model, we quantitatively compared the broad effects of simple genetic perturbations on the production of different VFs versus growth.

**Core set of growth-essential genes impact VFs.** To study the role of genes critical to both growth and VF synthesis, we compared the 116 genes predicted by mPA14 as essential for growth on SCFM to the genes essential for synthesis of VFs and found that 46 of the growth-essential genes are also essential for the production of at least one VF. These 46 genes critical to both biomass production and virulence are listed in Fig. 3 with their PseudoCAP category and function and a heatmap showing the affected VFs. The PseudoCAP category critical for the largest number of VFs is fatty acid and phospholipid metabolism, with seven genes predicted to be essential for the production of at least eight VFs in addition to biomass production. Additionally, several *aro* operon genes contributing to aromatic amino acid synthesis are essential for the production of six VFs in the phenazine and siderophore families, while an array of genes involved in purine metabolism fully inhibit only the production of A-band-O-antigen. Ultimately, this analysis provides a novel list of genes ranked by their impact on virulence pathways in addition to growth inhibition, which may assist the design of therapeutics with broad impact on metabolic processes.

We then expanded our analysis to all genes in our model, plotting inhibition of each VF versus growth (Fig. 4). Each point in the resulting plots indicates the level of growth inhibition (*x* axis) and VF inhibition (*y* axis) relative to wild-type for a given *in silico* knockout. All data points are transparent such that a high density of data points results in an increase in colour intensity. Thus, the colour intensity at the origin of the plots indicates a high number of gene deletions that have no effect on production

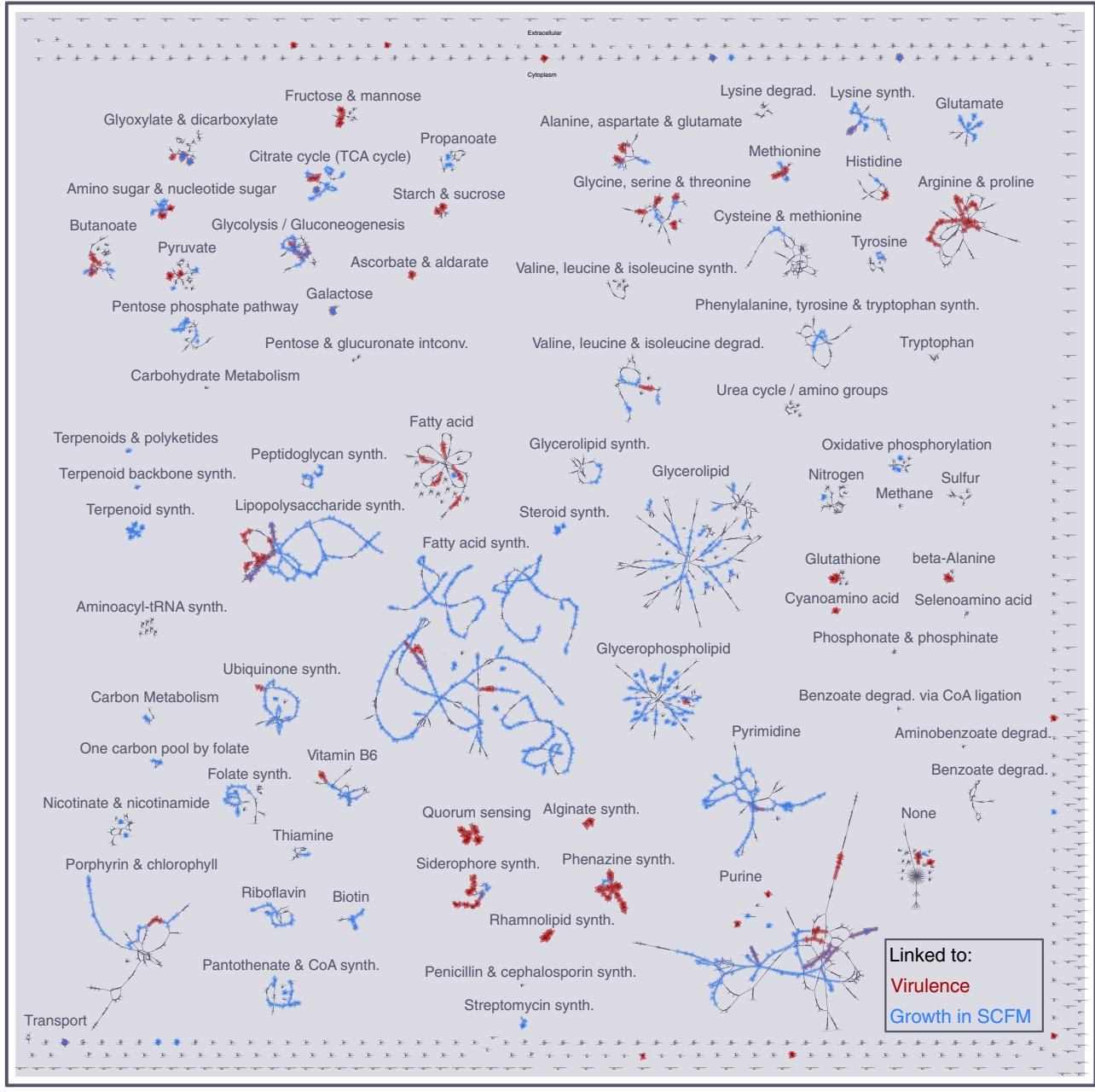

**Figure 2 | Visualization of experimental virulence-linked essential genes.** Distribution of virulence-linked genes and growth-essential genes from experiments in CF sputum visualized across all mPA14 reactions (grey) using MetDraw. Reactions associated with virulence-linked genes (as defined by the Pseudomonas Genome Database) are highlighted in red, and reactions associated with genes essential to growth in synthetic cystic fibrosis sputum are shown in blue. Purple reactions are associated with both virulence and growth essentiality. All reactions and metabolites are labelled with unique identifiers referenced in the model, visible at high magnification and text-searchable.

of either biomass or the indicated VF. Data points in the upper right corner of each plot represent genes essential to both VF production and growth, while data points arrayed between axes indicate the degree of biased impact on growth versus VF production by a given knockout.

This analysis enabled the identification of non-obvious relationships between growth and VF production. Unsurprisingly, most gene knockouts resulted in marginal or no growth defects, as indicated by data point clusters near the origin along the $x$ axis. This result was mirrored for VF synthesis, with most gene knockouts also resulting in marginal or no VF production defects. We hypothesized that VF synthesis would be less robust to perturbation as compared to growth because these compounds

rely on the catabolism of growth substrates prior to VF anabolism. We instead see that for several VFs, many genes essential for growth only partially inhibit synthesis when disrupted. The number of genes essential solely for the production of a given VF varies considerably, and is not always correlated with the complexity of the synthesis pathway or final compound. These results highlight critical differences in the degree of interconnectivity of VF synthesis and biomass production across the VFs, which we can evaluate mechanistically through the use of our computational model.

VFs that are less sensitive to genetic perturbations than biomass production may have a high degree of redundancy in their synthesis pathways. For example, relatively few genes impact

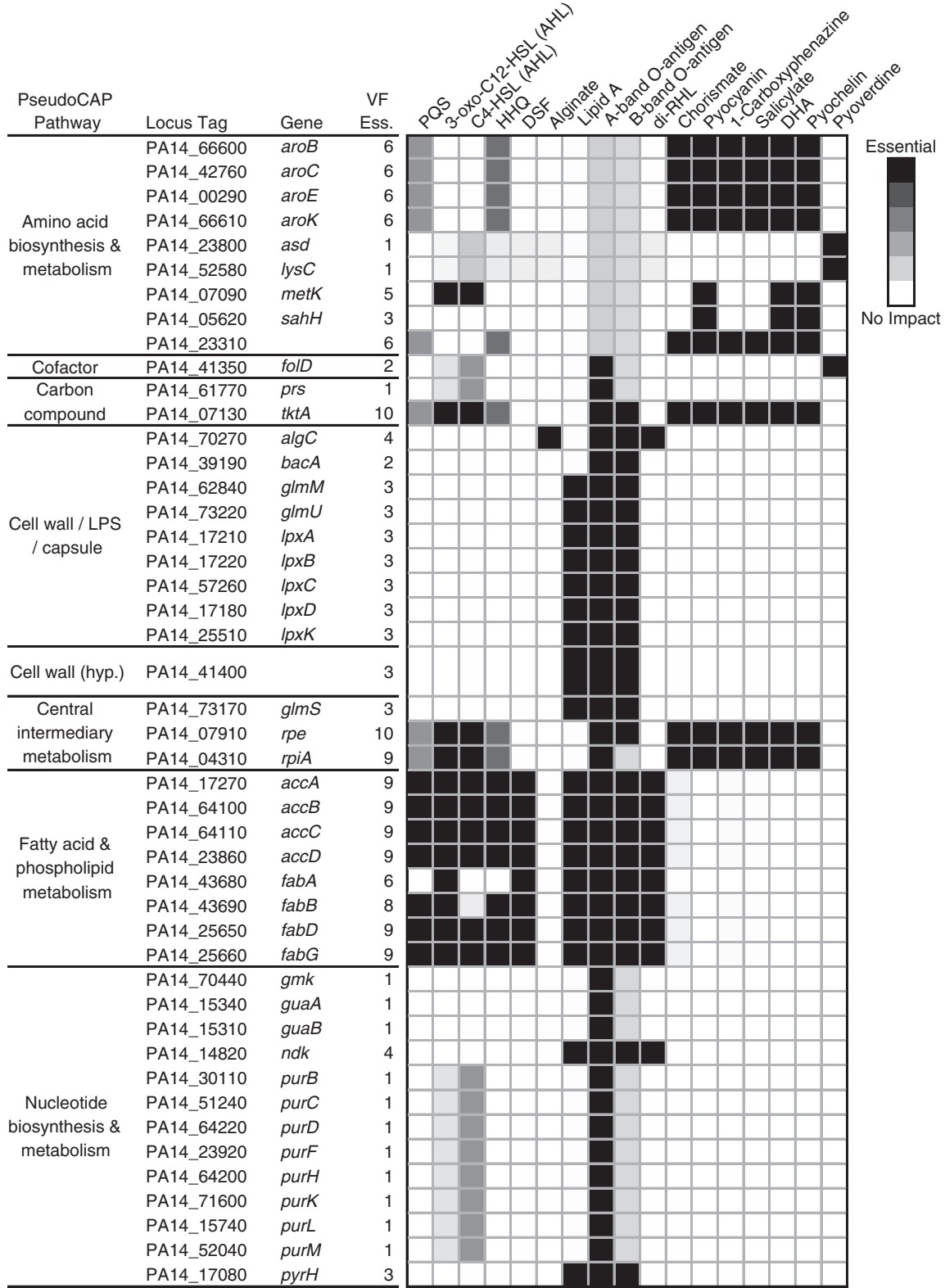

**Figure 3 | Genes essential for VF synthesis versus growth in SCFM.** The table lists the 46 genes essential for growth and production of at least one VF. Pathway assignment via PseudoCAP annotation and tabulated count of VFs for which the gene is essential are also included. Impact of a given gene's deletion is shown as white indicating 0% inhibition and black indicating 100% inhibition.

lipid A, chorismate and 1-carboxyphenazine production without also impacting growth and no gene is essential solely for the production of the respective VF. Instead, genes that are essential for VF production are also essential for growth, thus indicating the high level of integration of VF synthesis with the overall metabolism of *P. aeruginosa*. While this integration of VF and biomass synthesis is expected for lipid A given its presence in the biomass reaction in the model as an essential component, this

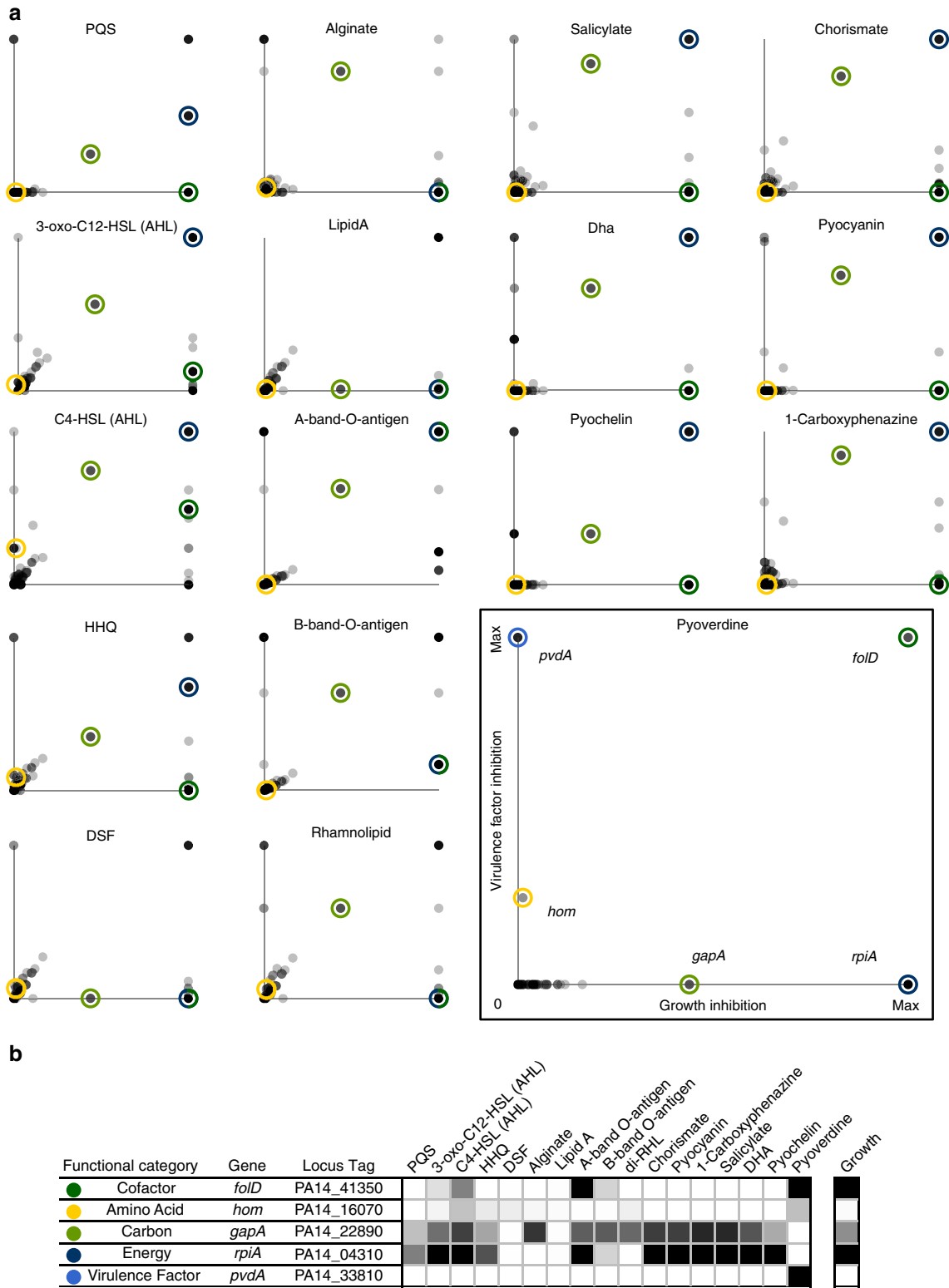

**Figure 4 | VF synthesis and growth interconnectivity.** (**a**) *In silico* gene knockouts were performed and the subsequent levels of biomass and VF production for each VF in the model were measured. The amount of growth inhibition was calculated by normalizing the knockout biomass production to the wild-type biomass production. Likewise, the amount of VF synthesis inhibition was calculated by normalizing the mutant level of VF production to the wild-type level of VF production. Each point indicates the growth inhibition (*x* axis) and VF inhibition (*y* axis) relative to wild-type for a given *in silico* knockout. All data points are transparent such that a high density of data points results in an increase in colour intensity. Coloured circles are used to indicate genes of interest as labelled in the pyoverdine example with yellow representing genes involved in amino acid metabolism, green carbohydrate metabolism, dark blue energy metabolism and light blue VF metabolism. (**b**) We highlight genes representing unique subtypes of impact on pyoverdine synthesis versus growth in a quantitative way that enables easy comparison of the activity of these genes across all VFs, with white indicating 0% inhibition and black indicating 100% inhibition.

was a surprising result for chorismate and 1-carboxyphenazine as we do not consider these essential components. Upon closer network inspection, we find alternative pathways for the production of chorismate and multiple isozymes for the synthesis of 1-carboxyphenazine. Both instances highlight redundancies in the network that reduce the occurrence of predicted essential genes unique to these two VFs.

Interestingly, B-band-O-antigen and A-band-O-antigen demonstrate the other extreme—all growth essential genes also impact the synthesis of both O-antigens to some extent. This case highlights the dependency of the production of these O-antigens on some of the biomass components themselves, namely lipid A. Since the O-antigens rely on the production of lipid A, all the genes that inhibit the synthesis of lipid A (and, thus, biomass) also inhibit the synthesis of the O-antigens. This result indicates the importance of biomass function formulation; here, we retain a standard list of components for consistency with other models, but more targeted analyses may improve upon addition and removal of less integral biomass components.

In contrast to these VFs, there were several VFs that displayed much less interconnectivity with the rest of the network. For example, relatively few genes maximally inhibit both growth and alginate production compared to the other VFs in the model. Most of the genes that are essential for alginate production have no impact on growth when the associated function is removed from the model, thus indicating that the synthesis pathway for alginate is less highly integrated into the metabolism of *P. aeruginosa*. VFs like alginate may be more peripheral to the general metabolic function of *P. aeruginosa* due to specialization. While *in vivo* studies have highlighted the importance of these metabolites in maintaining infections[31–34], here we systematically demonstrate pathway independence from essential metabolic function.

**Interconnectivity of synthesis of pyoverdine and biomass.** In addition to studying the interconnectivity of individual VF synthesis pathways, this analysis also enabled the investigation of the role of individual genes. While the disruption of some gene functions results in a similar response across the VFs, other gene function disruptions produce a highly varied response, suggesting that these genes play a unique role in the synthesis of each VF (example genes circled in Fig. 4). Using the VF pyoverdine as a reference, when the function of the gene *hom*, encoding for homoserine dehydrogenase, is removed, there is a very slight impact on growth and only marginal impact on each of the VFs, with pyoverdine synthesis demonstrating the most inhibition as a result of a *hom* knockout. Similarly, while a functional disruption of *pvdA*, which encodes for L-ornithine N5-oxygenase, maximally inhibits pyoverdine production, it has no impact on synthesis of the other VFs accounted for in mPA14.

In contrast, other simulated gene knockouts have more varied impact on VF inhibition. Functional disruption of *gapA*, which encodes for glyceraldehyde-3-phosphate dehydrogenase, has varied impact on the synthesis of VFs in the network, illustrated by preservation of pyoverdine production but near maximal impact on salicylate production. Similarly, while functional disruption of *rpiA*, which encodes for ribose-5-phosphate isomerase, again does not inhibit pyoverdine production, it does inhibit production of the AHLs incorporated into mPA14. Likewise, functionally disrupting *folD*, which encodes for 5,10-methylene-tetrahydrofolate dehydrogenase, maximally inhibits pyoverdine production and has no impact on PQS production. Thus, we can tease out the role of different genes on the synthesis of different VFs, with some simulated gene knockouts demonstrating consistent levels of inhibition

across all of the VFs and others demonstrating varied levels of inhibition.

**Experimental evaluation of pyoverdine mutants.** We chose to extend our investigation of the inhibition of pyoverdine synthesis because of the important role it has in iron scavenging and the tractability of experiments measuring pyoverdine production. In fluorescent Pseudomonads, pyoverdine is the main siderophore, a molecule that solubilizes iron for use by essential metabolic processes. It has been implicated in bacterial interactions in biofilms, it is essential for burn wound colonization, and it is upregulated in initial CF lung colonization[35–37]. Pyoverdine is also considered a 'public good' compound that is produced by select members of a community to benefit the whole. Thus, the inhibition of pyoverdine synthesis within the small group of producer cells may affect the whole community while reducing the possibility of acquisition and spread of resistance genes[38,39].

To interrogate the relationship between growth and pyoverdine synthesis, we identified gene function disruptions with varied impact on pyoverdine synthesis and growth as shown by the circled points of Fig. 4. We chose *pvdA* because it was predicted to be essential for pyoverdine production but not growth. Conversely, *rpiA* was chosen because it was predicted to be essential for growth but not for pyoverdine production. We chose *folD* because our model predicted it to be essential for both growth and pyoverdine production, and *hom* and *gapA* because of their predicted sub-inhibitory effects on pyoverdine production and growth, respectively. We then investigated the accuracy of these predictions with literature and experiments using available transposon mutants.

Mutants for both *folD* and *rpiA* were not present in the PA14 genome-wide transposon mutant library[26], suggesting that both these genes are indeed essential for growth of *P. aeruginosa*. Involved in the folate biosynthetic pathway, *folD* plays a critical upstream role in the synthesis of several compounds such as thymidine, purines and various amino acids. Studies have investigated *folD* as a potential therapeutic target to kill a variety of pathogens including *P. aeruginosa*[40–42]. Also important for purine synthesis, *rpiA* plays a critical role in the pentose phosphate pathway, converting D-Ribulose-5 to D-Ribose-5. Due to their growth essentiality, it is not feasible to study their role in VF synthesis experimentally—we instead use our computational model to offer unique insight. While mPA14 predicts that *rpiA* is not important in pyoverdine synthesis via a simulated knockout, it does predict that *folD* plays a crucial role, as evidenced by a simulated knockout resulting in total inhibition of pyoverdine synthesis. An analysis of pyoverdine synthesis precursors that cannot be produced after an *in silico folD* knockout in mPA14 highlights N5-formyl-N5-hydroxy-L-ornithine as the missing metabolite. This metabolite is not included in the much longer list of missing metabolites including purines that prevent biomass formation by the model. Thus, while *folD* may be essential for growth because of its role in purine synthesis, it appears to be essential for pyoverdine synthesis because of its role in amino acid metabolism. Understanding the metabolic interconnectivity of these genes provides insight into their potential impact on multiple systems if targeted therapeutically and we are able to determine the role of growth essential genes in VF synthesis which would otherwise be intractable.

Using transposon mutants of *pvdA*, *hom* and *gapA* from the PA14 genome-wide transposon mutant library[26], we performed absorbance-based assays of pyoverdine production and growth in SCFM as described in the methods. The extent of growth and pyoverdine production (normalized to growth) for wild-type

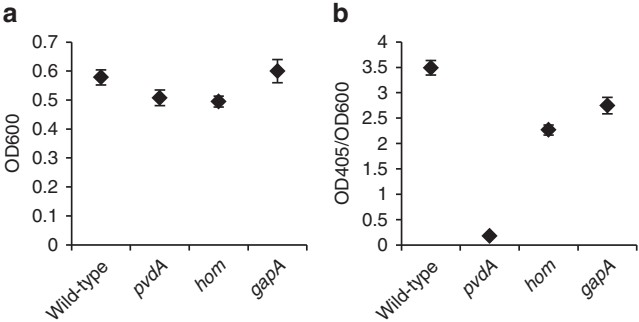

**Figure 5 | Pyoverdine synthesis capabilities *in vitro* on SCFM.** PA14 wild-type and *pvdA*, *hom* and *gapA* PA14 mutants were grown to stationary phase in SCFM and growth was measured using OD600 (**a**). Subsequently, the supernatants were isolated and the OD405 of each condition's supernatant was measured as a proxy for pyoverdine levels. The OD405 was divided by the OD600 of the culture in order to normalize pyoverdine production to growth (**b**). Error bars indicate s.d. among five biological replicates.

PA14 and each mutant strain are shown in Fig. 5a and b, respectively. As expected, the *pvdA* mutant showed markedly decreased pyoverdine production compared to wild-type, as it is an established pyoverdine assay control directly connected to the pyoverdine synthesis pathway. Interestingly, the *pvdA* mutant resulted in a minor growth defect relative to wild-type. This result could indicate that the lack of pyoverdine, and thus lack of access to iron, somewhat hindered the growth of the *pvdA* mutant. The *hom* mutant displayed a marginal growth defect and an approximately 1.5-fold decrease in pyoverdine production relative to wild-type. These results are consistent with model predictions that a *hom* knockout would result in slight growth inhibition and significant, but not total, inhibition of pyoverdine synthesis. Homoserine dehydrogenase, the gene product of *hom*, catalyses the reaction converting L-homoserine to L-aspartate 4-semialdehyde which then gets converted to L-2,4-diaminobutryate, an important precursor of pyoverdine[43]. Thus, a mutated *hom* limits the production of L-aspartate 4-semialdehyde, creating a bottleneck in pyoverdine synthesis. Because, L-aspartate 4-semialdehyde is not a growth-essential metabolite, both *in vitro* and *in silico*, targeting it may specifically prevent pyoverdine production without strong growth-based resistance selection. Unlike the *pvdA* and *hom* mutants, the *gapA* mutant did not exhibit a growth defect, disagreeing with our model prediction of an approximately 50% reduction in growth. Additionally, while we predicted that a *gapA* knockout would not impact pyoverdine production, we observed that the *gapA* mutant did indeed reduce pyoverdine synthesis, albeit to a lesser extent than the *pvdA* and *hom* mutants. *gapA* encodes for the enzyme glyceraldehyde-3-phosphate dehydrogenase, which catalyses the conversion of glyceraldehyde-3-phosphate to 1,3,-biphospho-D-glycerate, a central reaction in glycolysis. The discrepancy between our model predictions and experimental results for both growth and pyoverdine synthesis identifies a 'gap' in our knowledge regarding the function of glyceraldehyde-3-phosphate dehydrogenase in the overall metabolic network. Thus, we experimentally tested model predictions regarding genes in both growth and VF synthesis to tease out the role of genes upstream of pyoverdine synthesis and identify 'gaps' in our current understanding of *P. aeruginosa* metabolism.

## Discussion

We utilized a novel approach to systematically evaluate the contribution of metabolic genes to the synthesis of factors critical to virulence as well as growth at genome-scale using metabolic models. A new GEM for *P. aeruginosa* PA14 and an updated GEM for strain PAO1 were curated using single transposon insertion mutant data, virulence-linked gene databases, substrate utilization data, updated genome annotations and recent literature. Using our PA14 model, we contextualized the PGD database of virulence-linked genes that were identified as essential for growth in Tn-seq screens, and then identified a core set of metabolic genes that were necessary for both growth and the synthesis of at least one VF. Subsequent analyses mapped the metabolic interconnections between growth and the synthesis of individual VFs, using a case study of pyoverdine to demonstrate model utility in teasing out the role of individual genes with regards to both growth and VF production. Our work enhances understanding of relationships between VF synthesis and growth, which is challenging to elucidate with experimental approaches. By quantifying the impact of genetic targets on growth versus virulence using a mechanistic model, we contribute novel insights for the design of therapeutic strategies that account for potential resistance development.

The rapid fitness screening enabled by transposon insertion mutants have produced valuable insights into gene function in different environments[44], but the genome-scale data sets can be difficult to interpret. Signature-tagged mutagenesis screens in infection models have identified virulence-linked genes, but require a highly accurate replication of growth conditions *in vitro* for a baseline of effective comparison with infection[45,46]. Recently, Tn-seq has been used to measure *in vivo* gene fitness[44], but transposon insertion coverage, interpretation of Tn-seq results and essential gene identification are difficult to replicate across studies. Our model provides important nuance when considering the true definition of an 'essential' gene and survival fitness in varied environmental conditions; genes can be classified in a quantitative manner instead of a binary 'essential/nonessential' categorization. Thus, the high degree of variability in the number and roles of genes identified as critical for fitness even in different Tn-seq studies of the same growth environment can be elucidated when paired with mechanistic modelling. While we focused our efforts on mapping the distribution of virulence-linked genes in the data sets in an effort to understand their impact without the pressure of survival in a host, there is a rich opportunity to expand this comparison to the distribution of all metabolic genes in future work.

Our analysis provides an important expansion of genes to consider when studying VF synthesis during adaptation. We identify 46 genes as critical for the production of up to 10 of the 17 assessed VFs as well as biomass, which represent a novel core set of metabolic functions integral to the development of infection by *P. aeruginosa*. Long-term adaptation may result in altered virulence capabilities due to accumulated mutations in these genes as well as regulatory genes and genes linked directly to product synthesis[47]. When we expand our analysis to all model genes, we can then group genes by their functional impact on growth, VF synthesis or a combination of roles highlighting a higher degree of connection. This novel analysis provides testable hypotheses regarding the contribution of a gene to a given synthesis task. While we focus on experimental assessment of uniquely categorized genes in our pyoverdine analysis, this approach can be expanded to the other VFs. The analysis also demonstrates the flexibility of *P. aeruginosa* metabolism in a substrate-rich environment; the impact of competition for resources and the role of auxotrophy in evolving strains can be compared using predictions of differing optimal growth and production levels. Thus, further mapping of gene function may be enhanced by repeating the study in less complex and varied growth conditions.

Therapeutically targeting virulence-related pathways is an approach attracting much attention from a field struggling to find effective treatment for drug-resistant pathogens[1,48]. Quorum sensing inhibitors have been investigated through small molecule screening for a range of pathogens including *P. aeruginosa*[49–51] partially as a means of broad-spectrum anti-virulence treatment. A recent study showed that gallium-based quenching of extracellular siderophore activity successfully inhibited infection of caterpillars by *P. aeruginosa* while avoiding the development of resistance[52]. While inhibiting siderophore synthesis may increase resistance incidence in comparison to quenching, it will also enable pathogen-specific targeting of iron sequestration and avoid other systemic side effects (for example, radiation with respect to gallium). While these VFs are regulated by quorum sensing molecules, related signalling networks are complex; more direct routes of inhibition provide an efficient avenue for precision treatment. This study provides curated sets of potential targets for diminishing or preventing the production of a large array of VFs. Reducing experimental costs and time to identify targets while simultaneously elucidating the underlying mechanisms by which targets inhibit infection are major contributions of our models to effective development of new therapies.

Our quantitative analysis of metabolic gene contribution to both growth and virulence is the most comprehensive genome-scale computational screen to date of virulence-related metabolism. Concerns regarding resistance to growth-targeting antibiotics in the context of multi-drug treatments may benefit from incorporation of new therapeutics that target VF synthesis[53–55]. However, new proposals regarding sequential cycling of drugs with different mechanisms of action in an attempt to avoid drug resistance may favour drugs that inhibit VF production and growth simultaneously to maximize impact[56,57]. Our mechanistic modelling approach allows us to predict the graded contribution of a given target gene to growth versus virulence systems to aid in these treatment designs. Ultimately, our updated models are valuable tools for quantitatively assessing relationships that would be challenging to interrogate experimentally at genome-scale. Our experimental validation of model predictions indicates that our approach provides testable hypotheses of gene function that can be used to elucidate critical interactions that may inform development of 'resistance-resistant' therapeutics.

## Methods

### Metabolic network reconstruction.
Previously published iterations of *P. aeruginosa* PAO1 GEMs iMO1056 (ref. 19) and iMO1086 (ref. 20) were both used as resources during reconstruction efforts. iMO1056 was created using field-standard syntax consistent with many models in the BiGG database, while iMO1086 was built using the ToBiN platform which is not currently available[19,20]. Since these original models were published, the modelSEED has become a favoured draft reconstruction resource, and offers a comprehensive database of balanced reactions and metabolites referenced from KEGG and MetaCyc from which hundreds of draft models have been created for use within the modelling community[58]. In light of this, we used a draft conversion of iMO1056 to model SEED format as the starting point for our GEM update to enable consistency with our past *P. aeruginosa* models, improve annotation of model reactions and metabolites (KEGG IDs, E.C. numbers, pathway assignments) and enable easy comparison with a large collection of models created and curated by other groups[12,59–61]. Because the conversion was an automated step performed by the modelSEED in an early iteration of the SEED database, manual curation was performed to add additional species-specific reactions that did not successfully convert from the original iMO1056 model or were present in iMO1086 as well as to correct conversion errors in reaction stoichiometry, directionality and GPR assignments. Further updates to SEED reactions and metabolite names using the modelSEED database were implemented to ensure consistency, and a KEGG subsystem assignment was added to each reaction when possible[62].

The genomic contents of *P. aeruginosa* PAO1 and *P. aeruginosa* PA14 and two closely related pathogens from the *Burkholderia cepacia* complex were compared to assist development of new, reconciled GEMs for each strain from previously built models. *P. aeruginosa* PA14 is a primary clinical isolate that is used as a model strain due to its substantial virulence in a variety of hosts, while *P. aeruginosa* PAO1, a wound isolate, is the main reference strain of this species[48]. We used

*Burkholderia* species specifically because of their similarities to *Pseudomonas* as opportunistic Gram-negatives that also chronically infect cystic fibrosis patients and share similar virulence mechanisms. We also previously built and extensively curated GEMs for these species in modelSEED syntax as described further below, making them useful resources. *P. aeruginosa* PA14, *P. aeruginosa* PAO1, *Burkholderia cenocepacia* J2315 and *Burkholderia multivorans* ATCC17616 were compared using genome-scale reciprocal BLASTP with an E-value cutoff of 0.01 with no low-complexity filter using CLC Main Workbench (CLC bio, Aarhus, Denmark). Hits with E-values below 1E-40 were considered high confidence hits and automatically matched. Genes with hits that received a higher E-value score were manually evaluated based on predicted function, gene descriptions and PseudoCAP category (custom system/pathway annotations) on the PGD (ref. 24) before inclusion in the models in the few cases they were employed. There is a distinction, however, between confident gene matches between the organisms used, and utilization of genes annotated as hypothetical proteins in all species. We propose functions for a number of hypothetical proteins in the models, some of which are missing a specific functional annotation only in PA14 versus PAO1, and some of which are hypothetical proteins in both strains which we have utilized in the models based on functional domain associations and other predictions provided by PGD annotations, literature and manual curation based on BLAST results against other species. Many of these hypothetical proteins are implemented in transport reactions, fatty acid and lipid pathways, and VF pathways. A table of these low-confidence gene assignments and hypothetical proteins to which we assigned functions in the model is provided in Supplementary Data 1.

Updated, strain-specific biomass formulas were created using a field standard approach that approximates the biomass composition by accounting for DNA, RNA, protein, cell wall components, lipids and organism-specific compounds whose production is required for growth[9,12]. This effort expanded the number of components considered necessary for growth according to an improved biomass formulation and an updated search of literature pertaining to *Pseudomonas* species. Additional *Pseudomonas*-specific requirements, such as preference for ubiquinone-9 versus ubiquinone-8 as a key cofactor in respiration[63], were implemented. More specific lipids were implemented using recent studies from literature and as enabled by the expanded lipid reactions used in modelSEED draft reconstructions. Specifically, while iMO1056 and iMO1086 accounted for simple representations of cardiolipin, phosphatidylethanolamine, phosphatidylglycerol and phosphatidylserine, we implemented the specific saturated, unsaturated and cyclopropane fatty acids making up the full lipid profile of *P. aeruginosa* as described in the literature[64–67]. We provide further details of the new biomass formulations at the following website in conjunction with the model files in spreadsheet and SBML format (http://bme.virginia.edu/csbl/Downloads1-pseudomonas.html).

To fill gaps and improve predictions, additional model components were first derived from iMO1086 and recently published GEMs of *Burkholderia* species. We built on prior curation efforts while maintaining consistent modelSEED syntax to enable future cross-species comparisons and community modelling. If SEED reactions in *Burkholderia* models were not present in the new *Pseudomonas* SEED model, the high confidence BLASTP results were used in conjunction with the PGD and Burkholderia Genome Database[68] and literature to evaluate addition of these reactions. Many of the new reactions were added to increase the number of Biolog carbon sources accounted for in the *Pseudomonas* models (from only the PM1 substrate set to both PM1 and PM2a substrate sets); this effort was guided by previous work we performed for the highly catabolically flexible *Burkholderia*[12]. We also implemented new VF synthesis pathways using similar *Burkholderia* pathways as a guide. Other new reactions were added to expand lipid metabolism pathways using literature regarding *Pseudomonas*-specific lipid composition and the increased specificity of SEED reactions in this subsystem. Reactions implemented in the other well-curated SEED model available during our build work, *B. subtilis* iBsu1103, as well as reactions included in the MetaCyc and MetRxn databases were also used as resources[59,69,70]. PAO1 and PA14 genes categorized as linked to virulence via data from experimental studies incorporated into the PGD v3 (ref. 24) were specifically evaluated for inclusion in the models to expand clinically relevant functional prediction ability (Supplementary Data 2 and Methods—Screen and Database Assembly).

### Model validation.
Models were validated using new, comprehensive assessments of experimental data from genome-scale transposon libraries and carbon utilization screening. Similar data had been used with prior models, but unexpected discrepancies identified in comparisons between PAO1 and PA14 measurements motivated careful re-assessment of data sets and experimental confirmation of results.

Gene essentiality predictions were performed by *in silico* deletions of single genes while optimizing for production of biomass using flux balance analysis via the COBRA Toolbox[71]. Predicted essential genes were compared with a list of genes that were not successfully targeted by transposon insertions in both genome-scale transposon insertion libraries of *P. aeruginosa* PAO1 (ref. 25) and *P. aeruginosa* PA14 (ref. 26). By using genes lacking transposon insertions in both studies, which used different transposon systems and resulted in differing levels of insertion rate and genome coverage, we increased our confidence that these genes were truly essential for growth in rich media for *P. aeruginosa* strains. Curation

with essentiality data resulted in improved prediction accuracy of gene essentiality via curated GPR relationships as well as the addition of new components to the biomass formula.

Single carbon source catabolic ability of the strains was predicted by providing a single carbon source and salts to the model via exchange constraints and optimizing for biomass production using flux balance analysis[12]. Carbon utilization data were compiled from literature for both PAO1 and PA14, but discrepancies between studies motivated us to perform our own growth screens for both strains using Biolog phenotype arrays PM1 and PM2a. Growth curve screens were performed in triplicate using a microplate reader with shaking at 37 °C for 48 h. Curves were evaluated to identify substrates enabling growth versus no growth[12]. Results guided specific curation of catabolic pathways and expansion of transport systems included in the model to improve prediction accuracy.

**Screen and database assembly.** Information on virulence-linked genes was compiled from the PGD (ref. 24) (current as of February 2016) using the Annotations by Category tool that provides Virulence Factor Annotation lists for several strains. We used the lists for PAO1 and PA14, which provided 427 and 208 genes, respectively, which were culled by the PGD from experimental screens in many different infection models, the Virulence Factor Database, and the Victors database as indicated in Supplementary Data 2. The bias towards PAO1 is partly due to more screens and studies performed for PAO1 versus PA14 in the literature; however, 419 of these genes are present in both genomes. We assumed that many of the genes identified as virulence-linked in PAO1 could also be virulence-linked in PA14; however, virulence-linked genes truly active in only one strain would be of interest to track in future work; these genes must then have alternate functions in addition to a role in virulence. Nevertheless, building on the above assumption, we created a combined list of genes associated with virulence that included any genes noted in either list which were present in both genomes to which we then added strain-specific virulence genes. The resulting lists included 432 and 441 plausible virulence-linked genes for PAO1 and PA14, respectively; the strain in which each gene was originally classified as a VF is also indicated as well as its presence in each model in Supplementary Data 2.

For the Tn-seq-based essential gene analysis, we obtained gene essentiality data from eight recently published Tn-seq screens for PA14 and PAO1 in a variety of culturing conditions. These screens are listed in Table 2 and are identified by the name of the paper's senior author (Pier[16], Whiteley[17] and Manoil[18]), strain, and media condition. For the PA14 Pier dataset[16], we used the essential genes identified in Supplementary Table 1 of the original manuscript. Similarly, for the PAO1 and PA14 Whiteley data sets[17], we used the essential genes identified in Data set S1 and Data set S3 of the original manuscript, respectively. For the PAO1 Manoil data sets[18], we curated the 'General essential genes' identified in Data set S1 of the original manuscript to determine essential genes for each of the three media conditions studied: LB, sputum and pyruvate. Specifically, we applied a cutoff such that if a mutant for a particular gene failed to be generated in at least one of the independent transposon mutant pools for a particular media condition, that gene was deemed essential for that media condition. This approach does not take into account the location of the transposon insertion and, thus, may miss some essential genes. In the end, we obtained eight unique lists of essential genes for either PAO1 or PA14 in different media conditions based on the Pier, Whiteley and Manoil data sets. Once we obtained these lists of the essential genes identified in each screen, we compared them individually to the list of virulence-linked genes from the PGD database for either PA14 or PAO1 as appropriate. Genes that were in both a particular screen's essential list and the virulence-linked list were categorized as virulence-linked essential genes for that particular screen.

**Prediction of virulence-related production versus growth.** VF production capacity was first evaluated by optimizing the flux through an artificial 'demand' reaction for each virulence-related metabolite. Single gene deletions were implemented by identifying reactions for which a given gene was essential via the model's Boolean relationships and then constraining the flux through each of these reactions to zero. The effect of each of these deletions was evaluated by predicting production levels of each VF and biomass separately; resulting production levels lower than 0.001 were categorized as completely inhibitory (that is, the deleted gene is essential for production of that component). Production levels were normalized by maximum possible production of a component under wild-type conditions for comparison within VFs.

**Network visualization.** mPA14 was visualized using a command line implementation of MetDraw[72] that enables colour overlay which was then edited in Inkscape (https://inkscape.org/en/).

**Strains and growth conditions.** Wild-type strains of *P. aeruginosa* PAO1 and PA14 and PA14 single gene knock-out mutants from the PA14 non-redundant genome-scale transposon library[26] were grown in LB media supplemented with 15 µg ml$^{-1}$ gentamycin as necessary at 37 °C with aeration for liquid cultures.

**Pyoverdine assay.** To measure pyoverdine production, strains were grown in synthetic cystic fibrosis media[29] for 24 h in 50 ml flasks and the absorbance of culture supernatants was measured at 405 nm according to a previously published protocol[73]. All measurements were normalized to culture density as determined by the absorbance of the bacterial culture at 600 nm.

**Data availability.** The new metabolic network reconstructions for *P. aeruginosa* PAO1 and PA14, iPae1146 and iPau1129, respectively, are provided in spreadsheet format (Supplementary Data 3 and 4) that includes curation notes and SBML file format at our lab website (http://bme.virginia.edu/csbl/Downloads1-pseudomonas.html). The lab website also includes files detailing the development of strain-specific biomass formulas. Experimental data sets are available by request to the corresponding author.

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

## Acknowledgements

We thank Heath Damron, Mariette Barbier and Glynis Kolling for discussions about virulence factor assays as well as Edik Blais for discussions regarding modelling. Support for this project was provided by the National Institute of General Medical Sciences (GM088244 and GM108501) and the Whitaker Foundation.

## Author contributions

J.A.B., A.S.B. and J.A.P. conceived and designed the study. J.A.B. and A.S.B. reconstructed and curated the models with assistance from P.Y. and J.C.T. J.A.B. and A.S.B. conducted all simulations and analyses. A.S.B. performed the experiments. J.B.G. and L.J. provided commentary and revision for the manuscript written by J.A.B., A.S.B., and J.A.P.

## Additional information

**Competing financial interests:** The authors declare no competing financial interests.

