## [Peer review file · Nature Communications]

Reviewers' comments:

Reviewer #1 (Remarks to the Author):

The goal of "Reconstruction of the metabolic network of *Pseudomonas aeruginosa* to interrogate virulence factor synthesis," is to introduce an updated metabolic model of the PAO1 metabolic network, a newly constructed metabolic model of PA14, and establish a new type of analysis which uses these in situ metabolic models to systematically investigate the impact of metabolism on microbial virulence.

The study of bacterial virulence in the context of bacterial metabolomics is one that has significant potential to reveal new insights into antibiotic treatment. Whole genome reconstruction models of the metabolome could identify treatments that effectively inhibit virulence while preventing selection for resistance. This paper should be published in *Nature Communications* after minor revisions to address the points below

Major point:

1. The paper could explore knockouts that improve, not only impair, cell growth or virulence. Figure 4 illustrates the authors' analyses of the impact of gene knockouts on growth as well on 17 virulence factors as compared to the WT. From this analysis, it appears as though none of these knockouts lead to a faster growth or higher virulence compared to WT. Are there knockouts predicted to increase growth or virulence? If so, are these being constrained to the origin of these axes? If not, is there some bias in the model preventing these types of predictions? If these types of results are being constrained to the origin, then I wonder if this particular depiction of the analysis is too focused on bacterial growth limitation and virulence inhibition. I feel that this type of analysis could be more widely used if it predicts mutants that grow better or are more virulent under certain conditions. For instance, *Pseudomonas aeruginosa* could be engineered to overproduce compounds of commercial value such as rhamnolipids for biotechnology applications.

Minor points.

- In figure 4, each set of axes is using the same growth data for each simulated knock out mutant, therefore much of the data on the x-axes shown is redundant. There might be another way to show these data, perhaps as two separate graphs, one of the growth impact alone as a function of all mutants with a second plot looking at the virulence/growth impact ratio as a function of only the mutants of interest with respect to each virulence factor studied. However this change is not required and I leave that to the authors' discretion.
- The legend in figure 5 describes PA14 mutants in the genes *argD*, *lysC*, *pvdA*, and *pvdF* and I think this is incorrect according to the figure and the text which discusses PA14 mutants in *pvdA*, *hom* and *gapA*.
- I find the patterning used to denote metabolites and reactions in figures 1 and S1 difficult to read.
- Given the space available for figure S1, I would like to see the x-axis of the graph expanded to make the graph more readable.
- Line 470 I believe has a typo where there is a comma in place of "impact" and lipid is incorrectly capitalized.

Reviewer #2 (Remarks to the Author):

In their paper Bartel et al. greatly expand upon the available genome-scale models of metabolism in *Pseudomonas aeruginosa* (PA) strain PAO1 and develop a new model for PA14. They then use the models and experimental data to examine the link between production of virulence factors to cellular metabolism and growth. While the goal of finding new metabolic targets related to virulence that do not effect cellular growth are laudable; the authors have not specifically identified any unique target. I'm also confused as to why they needed two models when in the end they combined the experimental data for each strain into one dataset. The authors also make a number of assumptions that could greatly alter the accuracy of their predictions.

Major points:

My biggest problem is with the fact that the experimental essentially tests for virulence genes were conducted in an environment that did not necessitate generation of virulence factors. That authors admit to this and try to argue around it but to me this is a major flaw in the study.

When adding new reactions into the models, the E cutoff value of 0.01 is VERY high. The authors need to provide description in supplementary materials about what fraction of newly added reactions have high E-values and in such cases were any genes assigned to phenotypically essential reactions. Given that the authors are comparing number of essential genes between growth and virulence related scenarios, assigning GPRs with low confidence could drastically lower the accuracy of predicted outcomes.

The authors need to provide a better explanation of why the PA models were gap-filled with Burkholderia reactions? If there is no specific reason then why not use all the other curated pathogen models? Also, they need to give further details of selection criteria for including these genes in the PA models. Why were they not initially included in the Oberhardt model?

The authors based on intuition more than double the number of experimentally identified essential virulence-related genes in strain PA14 (208 to more than 419). This is a major expansion that requires some experimental validation.

The membrane compositions of pathogens have been known to drastically change under different conditions. How would similar changes in PA, affect the reported results in this paper?

I am troubled by the authors' broad use of the term "essential". Essentiality is condition dependent but it seems that for the case of Tn-Seq data the authors decided to pool genes from all different growth conditions into one list. I don't think this is a sound practice.

Figure 1: I'm confused about how the new model includes 90 new genes but a 172 new E.C. numbers. Were a lot of old Oberhardt designations changed? This has not been detailed in the paper.

Line 349: What were the conditions for which the PGD virulence linked genes identified? If

they are not similar to conditions for which Tn-Seq results were collected then the authors are comparing apples and oranges. The authors' admit to this in the next paragraph.

Minor points:

Line 159: the term pseudoCap is used without any description or reference to clarify what it means or where it comes from.

Lines 232-238: It is very confusing when the authors suddenly start using terms like Pier datasets without any reference so the reader knows that they are talking about the datasets they referenced in the previous sentence, specially since they are using the name of the corresponding author and the Nat com format eliminates these names in the reference section.

Reviewer #3 (Remarks to the Author):

The authors present a primarily bio-informatic analysis of the metabolic connections between virulence factor production and growth across different lab growth environments, for PAO1 and PA14.

The topic is interesting and integrating virulence factors into metabolic models is an important goal.

my main concern is that the motivations and implications of the work are not adequately developed. Is there a hypothesis? The goal of slowing resistance to new drugs is presented early on, but in a manner that is self-conflicting (e.g. lines 61-62 versus 68-69: if VF = fitness in host, then targeting VF = targeting fitness, and therefore a driver for resistance).

why target genes that are central to VF and growth? this is not clear to me. I also think that other causal connections are being missed - many VFs are turned on in absence of growth, and contribute to subsequent growth via nutrient acquisition (so VFs become drivers of growth). Of course this is a big layer of additional extracellular complexity - but it is central to the issue of infection / damage control.

We appreciate the thoughtful comments from the reviewers. Below we have kept the original reviewer comments in **black** and our responses are in **blue**.

Reviewer #1 (Remarks to the Author):

The goal of "Reconstruction of the metabolic network of *Pseudomonas aeruginosa* to interrogate virulence factor synthesis," is to introduce an updated metabolic model of the PAO1 metabolic network, a newly constructed metabolic model of PA14, and establish a new type of analysis which uses these in situ metabolic models to systematically investigate the impact of metabolism on microbial virulence.

The study of bacterial virulence in the context of bacterial metabolomics is one that has significant potential to reveal new insights into antibiotic treatment. Whole genome reconstruction models of the metabolome could identify treatments that effectively inhibit virulence while preventing selection for resistance. This paper should be published in Nature Communications after minor revisions to address the points below

Major point:

1. The paper could explore knockouts that improve, not only impair, cell growth or virulence. Figure 4 illustrates the authors' analyses of the impact of gene knockouts on growth as well on 17 virulence factors as compared to the WT. From this analysis, it appears as though none of these knockouts lead to a faster growth or higher virulence compared to WT. Are there knockouts predicted to increase growth or virulence? If so, are these being constrained to the origin of these axes? If not, is there some bias in the model preventing these types of predictions? If these types of results are being constrained to the origin, then I wonder if this particular depiction of the analysis is too focused on bacterial growth limitation and virulence inhibition. I feel that this type of analysis could be more widely used if it predicts mutants that grow better or are more virulent under certain conditions. For instance, *Pseudomonas aeruginosa* could be engineered to overproduce compounds of commercial value such as rhamnolipids for biotechnology applications.

While it is possible to predict gain of function due to component knockouts with metabolic network models, this will only occur in a highly constrained model using, for example, constraints on network activity based on gene expression. Knockouts might then prevent a constraint from being applied, enabling an alternate and more efficient route of production. However, the model does allow evaluation of optimal production under the same genotype in different media conditions. We've therefore added a comment about the future potential of this approach in the discussion and appreciate the reviewer's highlight of relevant applications.

Minor points.

- In figure 4, each set of axes is using the same growth data for each simulated knock out mutant, therefore much of the data on the x-axes shown is redundant. There might be another way to show these data, perhaps as two separate graphs, one of the growth impact alone as a function of all mutants with a second plot looking at the virulence/growth impact ratio as a function of only the mutants of interest with respect to each virulence factor studied. However this change is not required and I leave that to the authors' discretion.

We appreciate the reviewer's concerns, but combining the figures would reduce the ability to clearly see the diversity of responses for each individual virulence factor that we then discuss in the text. We designed this figure as such because we wanted to specifically highlight the relationship between individual virulence factors and growth.

- The legend in figure 5 describes PA14 mutants in the genes *argD*, *lysC*, *pvdA*, and *pvdF* and I think this is incorrect according to the figure and the text which discusses PA14 mutants in *pvdA*, *hom* and *gapA*.

We've corrected the legend for figure 5.

- I find the patterning used to denote metabolites and reactions in figures 1 and S1 difficult to read.

We've made the suggested corrections in figures 1 and S1.

- Given the space available for figure S1, I would like to see the x-axis of the graph expanded to make the graph more readable.

We've made the suggested corrections in figure S1.

- Line 470 I believe has a typo where there is a comma in place of "impact" and lipid is incorrectly capitalized.

We've made the suggested corrections at lines 519-520.

Reviewer #2 (Remarks to the Author):

In their paper Bartel et al. greatly expand upon the available genome-scale models of metabolism in *Pseudomonas aeruginosa* (PA) strain PAO1 and develop a new model for PA14. They then use the models and experimental data to examine the link between production of virulence factors to cellular metabolism and growth. While the goal of finding new metabolic targets related to virulence that do not effect cellular growth are laudable; the authors have not

specifically identified any unique target. I'm also confused as to why they needed two models when in the end they combined the experimental data for each strain into one dataset. The authors also make a number of assumptions that could greatly alter the accuracy of their predictions.

Major points:

My biggest problem is with the fact that the experimental essentially tests for virulence genes were conducted in an environment that did not necessitate generation of virulence factors. That authors admit to this and try to argue around it but to me this is a major flaw in the study.

As noted by the reviewer, we concede in our manuscript that our investigation of virulence-linked gene importance in the Tn-seq screens is hampered by lack of explicit pressure for infection-based virulence factors. However, we initiated this analysis to interrogate the roles of 'virulence'-linked genes in general metabolism because literature evidence is growing that virulence-linked genes often have important and alternate roles to virulence. We have edited and streamlined the manuscript to more clearly relate how we are comparing genes already identified as important to infection in prior *in vitro* and *in silico* studies to genes that appear essential in differing growth states to develop a metabolism-linked subset of virulence genes.

When adding new reactions into the models, the E cutoff value of 0.01 is VERY high. The authors need to provide description in supplementary materials about what fraction of newly added reactions have high E-values and in such cases were any genes assigned to phenotypically essential reactions. Given that the authors are comparing number of essential genes between growth and virulence related scenarios, assigning GPRs with low confidence could drastically lower the accuracy of predicted outcomes.

The reviewer is correct that using such a high E-value cutoff in an automatic transfer of reactions would be very unwise. We appreciate the comment, which highlighted the need for improved clarity of the related methods section. We used the bi-directional BLAST results as a reference during curation, but in essence PAO1 and PA14 metabolic genes that are well annotated are nearly identical in content, and these genes make up the bulk of the models. As briefly described in the methods paragraph prior to the 'Genome comparisons' section, we manually curated reaction additions to the new SEED model versus iMO1056 because the new SEED syntax was not easy to automatically compare with the old iMO1056 syntax, some of which was customized versus BiGG syntax, as well as wholly independent and isolated iMO1086 syntax. There was no bulk addition of model content on the basis of the BLASTed genomes using this 0.01 cutoff; these results were instead a starting point used in conjunction with PGD annotations, literature, etc, to determine the addition of new reactions; we were particularly careful about the few additions of reactions linked to genes with an E-value higher than $1e-40$. We have added a

clearer description in the methods that includes this information and further details of low confidence additions to the model in SupplementalTable1.xlsx

The authors need to provide a better explanation of why the PA models were gap-filled with *Burkholderia* reactions? If there is no specific reason then why not use all the other curated pathogen models? Also, they need to give further details of selection criteria for including these genes in the PA models. Why were they not initially included in the Oberhardt model?

The *Pseudomonas* draft models were built using the SEED database, and then we made further manual additions and corrections based on the curated models with which we had a high degree of confidence. We used the *Burkholderia* models and the *B. subtilis* model specifically as main references because they were the only curated SEED-based models at the time of the reconstruction work. *Burkholderia* is also an opportunistic Gram-negative pathogen with similar virulence factor synthesis pathways to *P. aeruginosa* and causes similar infections in cystic fibrosis patients. Some of the genes added were to expand virulence factor pathways while others were isozymes/gene duplications. Many genes were also added in the expansion of pathways that enable the evaluation of more Biolog carbon sources (iMO1056 was only validated using PM1 carbon sources from Biolog, and we included PM1 and PM2a), including a substantial effort to add more plausible genes connected to transporters. The lowest confidence genes that we added were those annotated as short-chain dehydrogenases active in lipid pathways, as there are many variant copies and usually low specificity regarding function. We used as many functionally specific genes with literature references as possible as shown in the model notes, but there are still generic dehydrogenases present. We had previously performed these tasks for the *Burkholderia* models, so they were a convenient starting reference. We have added a description of this effort to the methods; there are extensive notes by reaction included in the model spreadsheet files detailing our curation decisions (including notations where low-E value matches were used). We have also added a table of putative and hypothetical proteins and 7 proteins with alternate functions used to gapfill in SupplementalTable1.xlsx to address the concerns in this and the previous review comment regarding low confidence genes.

The authors based on intuition more than double the number of experimentally identified essential virulence-related genes in strain PA14 (208 to more than 419). This is a major expansion that requires some experimental validation.

We have edited the description of our use of the virulence-linked gene list to emphasize our interest in the potential multifunctionality of the virulence genes that we examine in the context of Tn-seq screens and model curation and prediction. We felt the combined list was appropriate given that a key thrust of our study was to evaluate the role of ‘virulence’ genes in metabolism as there are active questions in the field regarding the classification of these genes given discoveries of additional functionality. As most of the genes in the PAO1 list are present in the PA14 genome, and these strains are very similar genetically, we don’t feel it is a stretch to assume that

the roles of the genes may be similar in each strain; however, differences between the function of the same gene in different strain backgrounds indicates that the multifunctionality we are interested in may indeed be active.

Because we also used this list as a guide for our curation efforts, our goal was to provide a comprehensive resource where any genes that might be involved in virulence were accounted for and annotated in both models. Thus, we included our final set of virulence-linked genes and distinguished between their original annotation as a VF in PAO1, PA14, or both strains versus their inclusion in each model in SupplementalTable2.xlsx (Virulence-linked genes sheet). We have added information in the methods describing the virulence-linked gene database further.

The membrane compositions of pathogens have been known to drastically change under different conditions. How would similar changes in PA, affect the reported results in this paper?

While altered compositions might shift the balance of fluxes in lipid metabolism reactions, knocking out a gene is likely to have a much greater impact on model-predicted phenotype, and we are comparing WT predictions to knock-out predictions without varying anything else in the base model. We could repeat this knockout screen while iteratively altering the membrane composition, but the relative effect of most knockouts would likely not change. However, this is the first iteration of the *Pseudomonas* model that actually enables this analysis given our improvements to the lipid composition pathways, and is an interesting idea for a future study where we have appropriate experimental data to integrate. We have added further explanation of the lipid pathway improvements to the ‘Biomass’ methods section and are adding the detailed biomass formulation spreadsheets we used to the web page where the model files can be accessed.

I am troubled by the authors' broad use of the term "essential". Essentiality is condition dependent but it seems that for the case of Tn-Seq data the authors decided to pool genes from all different growth conditions into one list. I don't think this is a sound practice.

We appreciate the reviewer's justified concern about pooling Tn-Seq gene hits. We were careful not to pool potentially essential genes from different growth conditions/studies into a single list, and instead compared each screen's proposed essential genes to the corresponding PAO1 or PA14 virulence-linked genes list. This allowed us to compare how different screens identified not only different numbers of essential genes but also different numbers of virulence-linked essential genes. Furthermore, in the Figure 2 comparison map, we focused our analysis only on the Whiteley PA14 sputum study (Turner et al. PNAS. 2015.). We appreciate the reviewer's comment, which highlights that all of our 'lists' of different genes from different sources is quite difficult to follow in the manuscript presentation. We have added a clearer description in the methods of our approach to distinguish between these lists. We too believe that the term “essential” is often misused in the literature, and we hope that our study might help to clarify the

differences in “essentiality” among the various published studies. We’ve added a few relevant lines (Lines 719-722) in the discussion to add more clarity.

Figure 1: I'm confused about how the new model includes 90 new genes but a 172 new E.C. numbers. Were a lot of old Oberhardt designations changed? This has not been detailed in the paper.

The new models have many more E.C. numbers than the original model because we switched to using the SEED database as a source of reactions, and this database includes a much more comprehensive annotation of reactions and metabolites. This was part of the motivation for converting the syntax of the model using a larger maintained and curated public database that uses balanced reactions and links to many other databases (KEGG, MetaCyc, etc). We have added these details to the reconstruction methods section of the manuscript.

Line 349: What were the conditions for which the PGD virulence linked genes identified? If they are not similar to conditions for which Tn-Seq results were collected then the authors are comparing apples and oranges. The authors' admit to this in the next paragraph.

The conditions in which the PGD virulence-linked genes were variable, as were the conditions of the Tn-seq screens. We conducted the first comprehensive comparison of virulence-linked metabolic genes with respect to their roles in Tn-seq based fitness screens with the expectation that the fitness contribution of specific genes would vary, but that there would be a percentage of metabolic-linked virulence genes that were constant across all screens. Our analysis shows results to the contrary, which is why we thought this was worth including in the paper. We have added a clearer description of the PGD-based gene list to the methods to address this and previous comments.

Minor points:

Line 159: the term pseudoCap is used without any description or reference to clarify what it means or where it comes from.

We’ve added a descriptive reference at Lines 167-168 and 353-354.

Lines 232-238: It is very confusing when the authors suddenly start using terms like Pier datasets without any reference so the reader knows that they are talking about the datasets they referenced in the previous sentence, specially since they are using the name of the corresponding author and the Nat com format eliminates these names in the reference section.

We’ve included an explanation of our naming convention (see Lines 271-272).

Reviewer #3 (Remarks to the Author):

The authors present a primarily bio-informatic analysis of the metabolic connections between virulence factor production and growth across different lab growth environments, for PAO1 and PA14.

The topic is interesting and integrating virulence factors into metabolic models is an important goal.

my main concern is that the motivations and implications of the work are not adequately developed. Is there a hypothesis? The goal of slowing resistance to new drugs is presented early on, but in a manner that is self-conflicting (e.g. lines 61-62 versus 68-69: if VF = fitness in host, then targeting VF = targeting fitness, and therefore a driver for resistance).

We have streamlined our introduction and discussion to focus on our exploration of the critical interconnections between virulence and metabolism that may impact the design of therapeutic strategies that have the potential to avoid the development of resistance. As our work progressed, we developed further appreciation for the importance of understanding and mapping interconnections between virulence and metabolism which has not been sufficiently addressed in previous work; our goal was therefore to provide a new platform for this study on which we can further build relevant drug target identification endeavors that focus on mitigating the development of resistance and provide a means for a more comprehensive evaluation of fitness.

why target genes that are central to VF and growth? this is not clear to me. I also think that other causal connections are being missed - many VFs are turned on in absence of growth, and contribute to subsequent growth via nutrient acquisition (so VFs become drivers of growth). Of course this is a big layer of additional extracellular complexity - but it is central to the issue of infection / damage control.

We have edited the manuscript to further emphasize that we provide a novel means of classifying proposed drug targets in relation to their effects on virulence-linked metabolic pathways. However, as discussed above, we attempt to focus on mapping connections between virulence and metabolism rather than emphasizing specific target proposals. We felt this would be a more important contribution to the active debates in the field about which targets are more effective at avoiding resistance. There seems to be no clear consensus in the literature as to whether targeting only virulence-linked genes is an effective way of avoiding selection based on 'fitness' or if the focus should be on combinational strategies that increase the number of adaptations that must occur to induce resistance. We agree that VFs play a pivotal and underappreciated role in growth and intend that our study contribute new theories regarding which metabolic genes may enable this complex interplay between virulence and growth in the enhancement of fitness.

REVIEWERS' COMMENTS:

Reviewer #1 (Remarks to the Author):

This is a review for the revised version of “Reconstruction of the metabolic network of *Pseudomonas aeruginosa* to interrogate virulence factor synthesis”. I read the new version of the manuscript and the rebuttals to the three reviewers’ comments: I believe the paper is ready for Nature Communications. I explain my reasons below.

1. The paper presents a significant methodological advancement: an improved whole-genome metabolomics reconstruction of *Pseudomonas aeruginosa*. The new model expands previous models and should be a valuable resource for others in the *P. aeruginosa* community. The analysis of virulence factors is novel and relevant for our understanding of this widespread bacterial pathogen.
2. The authors addressed the points I made earlier mostly by direct replies in their rebuttal and made minor revisions to the text. This is fine, though: the paper did not need extensive reviews and my suggestions were mostly requests for clarification.
3. I read the other referee’s comments. The important issues raised are conceptual and concern fundamentals of the approach. Issues such as lenient E-value cutoffs are central to these approaches and I believe there are other places to voice these concerns, such as in critical review papers. The present study stands as a state-of-the-art example of the whole-genome reconstruction method that—despite potential shortcomings that may derive from its basic assumptions—has demonstrated value.

Therefore I recommend publication of this paper without additional comments.

Reviewer #2 (Remarks to the Author):

I appreciate the authors' detailed response to my comments. The changes they have made to the manuscript address my concerns. I recommend that the article be published in Nature Communications.

Reviewer #3 (Remarks to the Author):

the authors have improved the focus of the MS and I'm happy with the revision